# SWIR AOTF Imaging Spectrometer Based on Single-pixel Imaging

**DOI:** 10.3390/s19020390

**Published:** 2019-01-18

**Authors:** Huijie Zhao, Zefu Xu, Hongzhi Jiang, Guorui Jia

**Affiliations:** School of Instrumentation Science and Opto-electronics Engineering, Beihang University, Beijing 100191, China; gongxunbuaa@163.com (H.Z.); zefuxu@126.com (Z.X.)

**Keywords:** imaging spectrometer, acousto-optic tunable filter (AOTF), single-pixel imaging (SPI) technique

## Abstract

An acousto-optic tunable filter (AOTF) is a new type of mono-wavelength generator, and an AOTF imaging spectrometer can obtain spectral images of interest. However, due to the limitation of AOTF aperture and acceptance angle, the light passing through the AOTF imaging spectrometer is weak, especially in the short-wave infrared (SWIR) region. In weak light conditions, the noise of a non-deep cooling mercury cadmium telluride (MCT) detector is high compared to the camera response. Thus, effective spectral images cannot be obtained. In this study, the single-pixel imaging (SPI) technique was applied to the AOTF imaging spectrometer, which can obtain spectral images due to the short-focus lens that collects light into a small area. In our experiment, we proved that the irradiance of a short-focus system is much higher than that of a long-focus system in relation to the AOTF imaging spectrometer. Then, an SPI experimental setup was built to obtain spectral images in which traditional systems cannot obtain. This work provides an efficient way to detect spectral images from 1000 to 2200 nm.

## 1. Introduction

An acousto-optic tunable filter (AOTF) is an optical band-pass filter controlled by a radio frequency (RF) drive signal. Every single band is matched with each RF signal. The AOTF was developed in the late 1960s with the collinear AOTF of Harris and Wallace [1], when optical and acoustic beams propagate collinearly. Then, Chang [2] introduced a non-collinear AOTF that uses a large-angle aperture. Most early studies were based on the theoretical contributions of Chang. Yano and Watanabe [3] explained the principle of AOTF and established the commonly accepted momentum matching and phase match conditions.

In the spectral region, an AOTF spectrometer is a type of imaging spectrometer that is different from the traditional prism and grating. An AOTF is an excellent dispersion element with the advantages of stability and a no-sweep mechanism. At present, the AOTF spectrometer is applied to many fields, such as deep space [4,5,6], the spectrometric study of plants [7,8], atmospheric monitoring [9], and security [10].

In the AOTF spectrometer, all the optical elements are matched with the parameters of the AOTF and a camera. The most serious bottleneck of the spectrometer is the AOTF because the radiant energy is limited by the optical aperture and the acceptance angle of the AOTF; however, a large-aperture AOTF is difficult to produce [11]. Accordingly, the throughput of a high-spectral-resolution AOTF system is limited given the confines of the AOTF.

Generally, a spectral system requires a camera with high sensitivity—this is especially true of the AOTF spectrometer. The advent of focal plane arrays (FPAs), such as a charge-coupled device (CCD), offers less dark current at visible wavelengths. However, in a short-wave infrared (SWIR) region, the FPA is based on various materials, including indium antimonide (InSb), indium gallium arsenide (InGaAs), and mercury cadmium telluride (MCT) [12,13]. Among these materials, the InSb FPA exhibits the lowest quantum efficiency. Furthermore, the InGaAs FPA can only detect in the range of 900–1700 nm and requires the MCT FPA to obtain an image when detecting out of range. The MCT FPA that has low dark current is prohibitively expensive or unavailable; examples include Sofradir Corp’s deep cooling products or Rockwell’s HgCdTe detector array, which are used by the National Aeronautics and Space Administration [4]. The high-performance MCT FPA is more suitable for deep-space applications than ground-based applications. At present, MCT FPAs that are used in the SWIR region can operate at roughly 200 K with Peltier cooling [14]. However, the dark current of a SWIR-MCT FPA is more than that of the InGaAs FPA; we cannot obtain effective data, except for the dark current. A new material of the SWIR FPA that can detect images in the range of 900–2500 nm has been developed recently, known as InAs/GaSb. This material is also called T2SL, and its dark current is approximately the same as that of the SWIR-MCT FPA. Therefore, only InGaAs FPAs operate effectively and are easily accessible in the low light condition of SWIR. Most ground-based SWIR AOTF systems must use the InGaAs FPA, which can only detect in a range of 900–1700 nm [15,16].

In contrast to the FPA, a single-element detector easily offers high sensitivity and detects added bands, such as the InGaAs single-element detector, even if these bands are unable to detect an image. For example, “Chang-e 3” a Chinese lunar probe, uses an InGaAs single-element detector as a detector of a non-imaging spectrometer [6]. Its spectral range is 899–2402 nm in the SWIR band. Moreover, a single-element detector covers a smaller area than an FPA, and all the energy is focused on an image plane; thus, an AOTF point spectrometer can avoid low light levels [17].

Furthermore, single-pixel imaging (SPI) is a novel technique for obtaining images by measuring coded signals. Many studies have been conducted on SPI in recent years, especially SPI by compressive sampling [18,19,20]. SPI characteristics include high sensitivity and low cost, and this technique is naturally suitable for imaging under low light conditions [21].

The low cost of photodetectors and the large operating bandwidth afforded by digital micromirror device (DMD) technology opens a range of alternative imaging solutions, such as hyperspectral imaging, particularly for imaging at wavelengths where CCD or Complementary Metal Oxide Semiconductor (CMOS) imaging technology is limited. Welsh et al. demonstrated a system that utilizes single-pixel photodetectors to produce a full-color high-quality image [22]. Meanwhile, many studies start to obtain spectral images by SPI [23,24,25,26,27]. However, the wavelength of their spectral images is in the visible region, while research in the SWIR region is lacking.

In this work, we integrate SPI and the AOTF spectrometer into one system. The AOTF spectrometer based on SPI collects more light per unit area than the pixel array approach. It is useful in reducing non-idealities of detectors, such as dark currents. The SWIR-MCT detector can be used to build SWIR AOTF SPI experiments and acquire spectral images in the range of 1000–2200 nm. Compared with the traditional SWIR AOTF spectrometer, SPI extends the spectral range to 2200 nm. This range is significant for classification applications; for example, water presents its characteristic spectral absorption peaks at 1.9 μm [4].

The remainder of this paper is organized as follows—the background of this work is explained in Section 2, the method is demonstrated in Section 3, the measurement system and experimental results are discussed in Section 4, and the discussion and conclusions drawn from this work are presented in Section 5.

## 2. Background

An acousto-optic (AO) device is developed by bonding an ultrasonic transducer to a suitable AO material. An ultrasonic transducer transfers RF signals to acoustic waves in an AO crystal. Then, a birefringent AO interaction diffracts different wavelengths of light (Figure 1). The AO device is considered an optical filter. The undiffracted light is interrupted by optical stops, and the diffracted light becomes a single-band image on the focal plane.

A simplified optical structure of the traditional AOTF imaging system is illustrated in Figure 2. In the system, the light passes through the AOTF, and then the AOTF (TeO_2_ spectral range: 900–2500 nm, spectral resolution: 10–20 nm) diffracts a narrow band into an imaging lens group and camera (InGaAs FPA produced by Xenics from Belgium, FPA size: 9.60 mm × 7.68 mm, wavelength: 900–1700 nm). The imaging lens group is designed to match with the AOTF and the FPA and can be regarded as a fixed singlet.

However, the aperture and acceptance angle of the AOTF are limited by various reasons, hence a high-performance detector is required. In the range of 900–1700 nm, the InGaAs detector can satisfy these requirements. In the range of 900–2200 nm, the SWIR-MCT is characterized by considerable noise. In a short integration time, the signals are considerably weak. However, the signals in a long integration time will be submerged by the dark current of the SWIR-MCT FPA, as displayed in Figure 3. 

## 3. Method

### 3.1. Setup of the AOTF Imaging Spectrometer Based on SPI

Owing to the difficulties in producing a large-aperture AOTF and the limitation of the camera, we applied SPI to solve the problem of the poor sensitivity of the existing AOTF imaging spectrometer. In this study, an experimental setup was designed (Figure 4). A spatial light modulator (SLM) modulated the light before the AOTF, and then Lens 1 focused on the SLM. Lens 2 collimated the light into the AOTF. A short-focus lens (focus length 8 mm) behind the AOTF was used to collect light.

Compared with pixel array imaging, SPI collected light into a small area. In this experiment, we selected an 8-mm lens to focus on an area approximately 0.8385 mm in diameter. Simultaneously, the diameter of the pixel array imaging area was 7.68 mm. The irradiance on the focal plane was calculated by the following equation:(1)$$E=\frac{d\phi }{dA}$$
where E is the irradiance, dA is the irradiated area, and dϕ is the radiation power through the AOTF. We assume that $E_{1}$ is the irradiance in a previous structure, and $E_{2}$ is the irradiance in a new structure. The ratio between $E_{1}$ and $E_{2}$ can be calculated using the following equation:(2)$$\frac{E_{2}}{E_{1}}=\frac{d\phi /dA_{2}}{d\phi /dA_{1}}=\frac{dA_{1}}{dA_{2}}=\frac{7.68\;\mathrm{mm}\times 7.68\;\mathrm{mm}}{0.8385\;\mathrm{mm}\times 0.8385\;\mathrm{mm}}\approx 83.9$$
That is, the irradiance of a short-focus system would be approximately 83.9 times higher than that of a long-focus system.

To acquire the spectral images, we adopted an SLM to modulate and reconstruct images by a specified algorithm. At present, the SLM is categorized into two typical types—a digital micromirror device (DMD) and liquid crystal on silicon (LCOS). A DMD indicates a higher transmission efficiency than an LCOS reflective spatial light modulator and therefore the former is effective in holding energy. Moreover, an LCOS SLM lacks developed products in the range of 1000–2200 nm. Thus, we selected a DMD produced by Texas Instruments (Dallas, TX, USA) in this system.

### 3.2. Principle of SPI with a Hadamard Mask Pattern

Image transformation is generally achieved through two methods that use different mask patterns, namely, Fourier and Hadamard basis patterns. Fourier basis patterns are grayscale (8-bit), whereas Hadamard basis patterns are binary. A DMD operates much slower in the grayscale mode than in the binary mode. Accordingly, Hadamard is a matrix that is naturally suitable for SPI systems based on a DMD [28].

In the process of image transformation (Figure 5), a short-focus lens collected light into a detector. The detector measured the signal $S_{p}$. The measurements of $S_{p}$ were real-valued, and the number of patterns was the same as that of the image pixels. The signal $S_{p}$ was associated with a Hadamard masking pattern H and was defined as follows:(3)$$\begin{array}{l} S_{p}=\sum_{j}\sum_{i}\left[H\left(i\times j-j+i,p\right)\times I\left(i,j\right)\right]+E_{p},\\ i=1,2,\cdots ,w;j=1,2,\cdots ,h;p=1,2,\cdots ,w\times h \end{array}$$
where I is the image plane that we wish to reconstruct, i and j index the *x*- and *y*-coordinates of the image plane, H is the Hadamard matrix with a size of wh×wh, p is the serial number of the acquisition, $E_{p}$ is the noise of detection, w is the width of the image, and h is the height of the image.

Owing to the dark current and read-out noise of the detector, the measurement of $S_{p}$ consisted of signals and noise. To reduce the series of noise, we obtained differential signals by showing each Hadamard mask immediately followed by its inverse (Figure 6). We supposed that ${S\prime }_{p}$ is the signal of the inverse coded image. Then, $S_{p}-{S\prime }_{p}$ is the difference in the measured intensities. This approach is an effective method for obtaining improved data [29].

After sampling, the image was reconstructed as:(4)$$I=H^{-1}\times \left(\begin{array}{l} \begin{array}{c} S_{1}-{S\prime }_{1}\\ S_{2}-{S\prime }_{2}\\ \vdots \\ \begin{array}{c} S_{p}-{S\prime }_{p}\\ \vdots \end{array} \end{array}\\ S_{w\times h}-{S\prime }_{w\times h} \end{array}\right)$$
where I is the image with a size of w×h and $H^{-1}$ is the inverse of the Hadamard matrix H.

### 3.3. Process of Grabbing Spectral Images

The passband of the AOTF imaging spectrometer was controlled by an RF drive, and the computer controlled the RF drive. Then, we designed a software program to ensure the sequence of the detector, drive, and DMD. The process was organized as follows:

Step 1: The detector was opened and preheated for approximately 30 min. This step was important for reducing the acquisition noise.

Step 2: The integration time of the detector was set to the same as the refresh cycle of the DMD. Otherwise, the state of the micromirror could not be determined at the end of the integration time, thereby resulting in fluctuations of measurements.

Step 3: An interesting band was selected, and the RF was tuned to a corresponding frequency. The response time of the AOTF was less than 1 ms.

Step 4: The object in the scene was adaptively located, and a Hadamard matrix H was generated.

Step 5: The pth column of H was selected as the masking pattern, and the signal $S_{p}$ was acquired. The column of H was reshaped to be the same size as the image because the column of H was a column vector.

Step 6: The masking pattern was immediately changed into its inverse, and the signal $S\prime_{p}$ was acquired. 

Step 7: Another column of H was selected, and Steps 5 and 6 were repeated until all the signals were acquired.

Step 8: The spectral image was reconstructed using Equation (4).

Step 9: Another interesting band was selected, and Steps 3–8 were repeated.

## 4. Experiments and Results

### 4.1. Camera Response Contrast between Long-focus Lens (Lens of the Traditional System) and Short-Focus Lens (Lens of the SPI System)

In the proposed method, we claimed that the irradiance of a short-focus system would be approximately 83.9 times higher than that of a long-focus system. Therefore, we measured the camera response with different focus lenses when the AOTF aperture was an aperture stop in the system. Figure 7 depicts the experiment setup, with the integral sphere as a uniform light source ranging from 400 nm to 2500 nm. We kept the stripe target before the integral sphere so the camera could collect images with different lenses.

The result of the experiment is shown in Figure 8. Obviously, the image collected by an 8-mm lens had a high contrast ratio. Meanwhile, we obtained a row of data from each image. The value of the bright pixel in the 8-mm image was about 16,000–18,000 and the value of the bright pixel in the 73-mm image was about 220–240. Therefore, the response of the 8-mm focus system was about 72–75 times larger than that of the 73-mm focus system. 

Due to the camera response, the irradiance was linear (Figure 9) when the number of pixels was less than 20,000. The irradiance of the short-focus system was also 72–75 times higher than that of the long-focus system.

When the target was a light source, we obtained signals with a long-focus lens. However, in practical applications, the intensity of reflected light is weak. Figure 10a,b display results obtained in weak light conditions. As Figure 10d shows, the signal was flooded by noise when the long-focus lens was adopted, but the short-focus lens obtained the signal (Figure 10c). 

Although a short-focus lens can collect light into a smaller area, the spatial resolution is greatly reduced. SPI can reconstruct high-resolution images with a short-focus lens. Moreover, it provides a low-cost way to obtain images. Therefore, SPI is used to obtain high-resolution images in the SWIR region.

### 4.2. SPI Imaging Spectrometer and Results of Different Objects

The real conditions of our experiment are depicted in Figure 11. All the elements were adjustable to regulate the optical path expediently. The object was placed approximately 1 m away from the DMD. The PC controlled the acquisition sequence between the AOTF drive, DMD, and detector.

A polarizer was used to suppress stray light caused by the non-diffractive light of the AOTF. The AOTF was placed in the plastic element that was designed to limit the field of view. For a better contrast, the detector was a 14-bit quantization SWIR-MCT FPA, which was also the detector in the traditional system. In addition, the FPA was convenient for adjusting the detector position and checking on the field. We calculated the sum of the effective area of the FPA center as a measurement signal. The AOTF and DMD in the experiment were the SWIR products. Table 1 lists several important parameters.

Certain results of this work are presented in Figure 12. We obtained several images of a typical case to assess the imaging quality of different objects. In this test, the driving RF of the AOTF was 49 MHz, and the band of images was approximately 1935 nm. The integration time of the detector was 8.333 ms, which matched the refresh frequency of the DMD. All the settings of the test were fixed, except for the target objects.

Figure 12a illustrates a white circle with a dark background. Figure 12b exhibits an oblique stripe. Compared with the horizontal and vertical edges displayed in Figure 12c, the oblique edges are obviously pixelated. Figure 12d presents a cut leaf of a real plant called *Epipremnum aureum*. Real plants exhibited a high reflectivity at the wavelength range of 750–1300 nm. However, the cut leaf was dark in the 1935 nm band, and the leaf reflectivity was low because H_2_O occupies an absorption peak in this band.

### 4.3. High- and Low-Resolution Images

The resolution of the images varied in accordance with the mask template, when SPI was adopted. From the different applications, we can obtain images of different resolutions. Therefore, in this experiment, we used 32 × 20 and 128 × 80 Hadamard masks as a comparable pair. As can be seen in the different resolutions (Figure 13), the resulting object was an oblique square pattern. The difference was clear because the edge of the 32 × 20 image was pixelated. However, the acquisition time was much longer in the high-resolution image than in the low-resolution image. 

### 4.4. Images of Different Bands

The spectral images of different bands from an imaging spectrometer are displayed in Figure 14. The target object was a leaf, and the background was white paper, which can be regarded as a standard reflectance.

The results revealed that a leaf exhibited high reflectance when the wavelength was less than 1400 nm, because the contrast ratio between the leaf and the paper it was located on was minimal. In the 1465 and 1935 nm bands, the leaf had less reflectance than in proximate bands because H_2_O occupies an absorption peak in these bands.

### 4.5. Experiment between Real and Fake Plants

In the actual application of imaging spectrometers, recognizing fake objects is an important feature. In several special bands, fake objects are obviously different from real objects. Therefore, we conducted a test between real and fake plants in typical passbands. 

Initially, the RF was adjusted to 49 MHz, and the passband was approximately 1935 nm. The target objects were a real leaf and a polyester fiber leaf. For improved illustration, a black background was placed behind the fake leaf, and a white background was placed behind the real leaf. The acquired images are depicted in Figure 15. The bright leaf in Figure 15d is the fake leaf, and the dark leaf in Figure 15b is the real leaf.

## 5. Discussions and Conclusions

The traditional AOTF imaging spectrometer cannot detect spectral images in the range of 1000–2200 nm by the non-deep cooling MCT FPA. In this work, a SWIR AOTF imaging spectrometer based on SPI was demonstrated to possess characteristics such as high sensitivity and adjustable spatial resolution. The spectrometer can obtain spectral images beyond 1700 nm and is used to detect the 1900 nm band, which is the feature band of H_2_O. In the experiment, the SWIR AOTF imaging spectrometer based on SPI acquired images of different objects, resolutions, and bands. The work provides an efficient way to detect spectral images in the range of 1000–2200 nm.

Several non-ideal behaviors were observed. The results indicate considerable noise, especially in the 1037 nm image, because the diffraction efficiency of the AOTF is low at 1037 nm and the noise has greater impact. In the future, instead of the FPA, an InGaAs single detector will be adopted as it has a lower dark current level and provides an economical way to obtain SWIR spectral images.

## Figures and Tables

**Figure 1 sensors-19-00390-f001:**
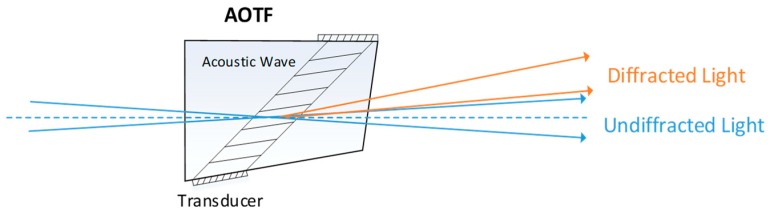
Principle of an acousto-optic tunable filter (AOTF) device. The frequency of the acoustic wave is changed by the AOTF drive and the wavelength of diffracted light changes together.

**Figure 2 sensors-19-00390-f002:**
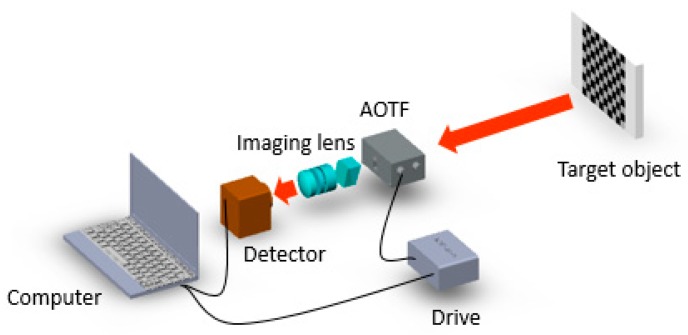
Simplified structure of the traditional short-wave infrared (SWIR) AOTF imaging spectrometer. The focal length of the imaging lens is 73 mm. The AOTF aperture is an aperture stop in the system. The AOTF and the detector are controlled by software. The target object is a checkerboard.

**Figure 3 sensors-19-00390-f003:**
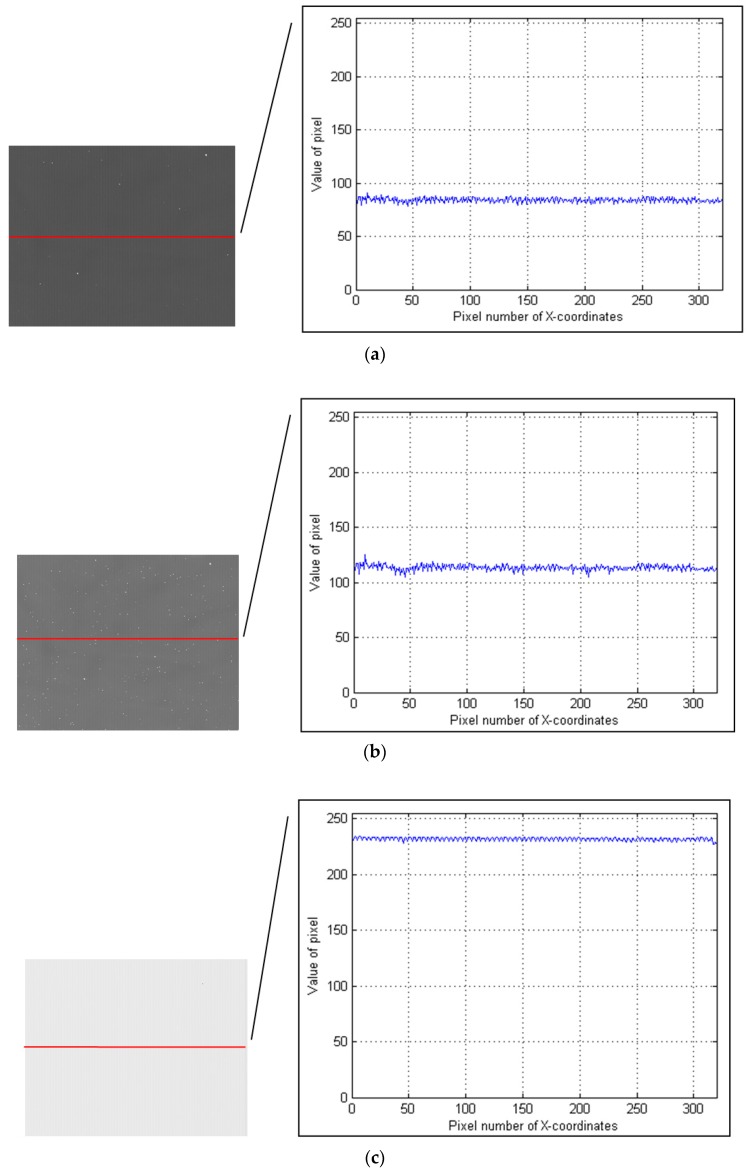
Spectral images acquired by the traditional SWIR AOTF imaging spectrometer (the detector is a non-deep cooling mercury cadmium telluride (MCT) focal plane array (FPA)). The curve on the right is the row of data contained in the red line on the left. The data here are the results of 8-bit quantization and 255 is the max number of pixels. The integration times of images (**a**–**c**) are 5, 10, and 20 ms, respectively. The target object is a checkerboard, but the signals will be submerged by a dark current no matter which integration time is adopted.

**Figure 4 sensors-19-00390-f004:**
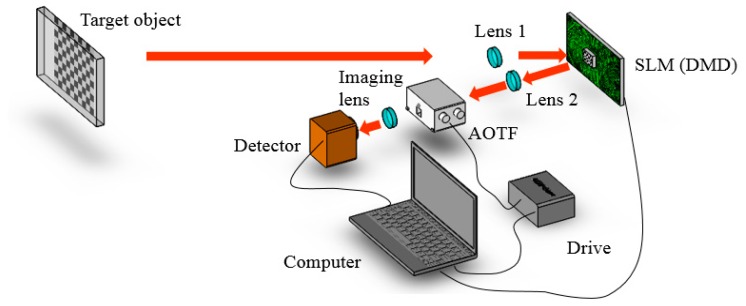
The setup of an AOTF imaging spectrometer based on single-pixel imaging (SPI). The focal length of the imaging lens was 8 mm. Lens 1 was focused on the digital micromirror device (DMD) and Lens 2 was the collimating lens. The DMD displayed the mask.

**Figure 5 sensors-19-00390-f005:**
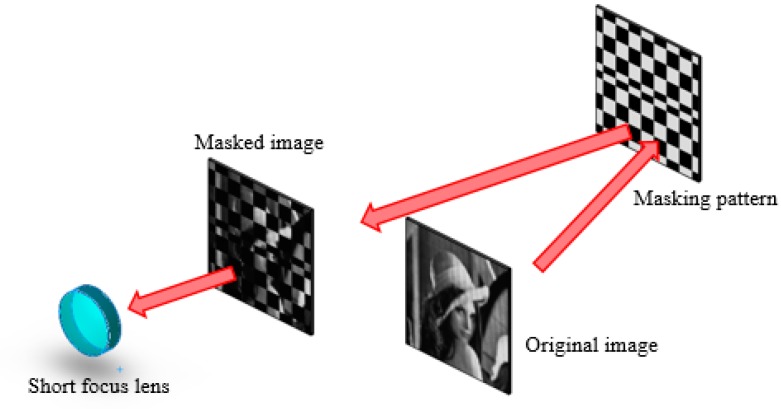
Process of image transformation. The short-focus lens collected the light into a small area on the focal plane. The focal length of the short-focus lens was 8 mm in this case.

**Figure 6 sensors-19-00390-f006:**
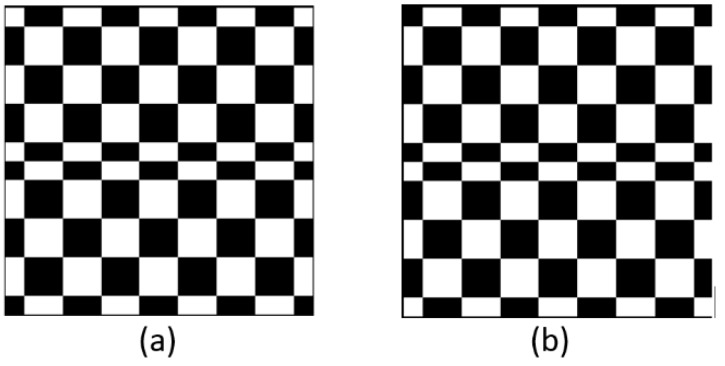
Original Hadamard mask and its inverse: (**a**) the original Hadamard mask and (**b**) the inverse of (**a**).

**Figure 7 sensors-19-00390-f007:**
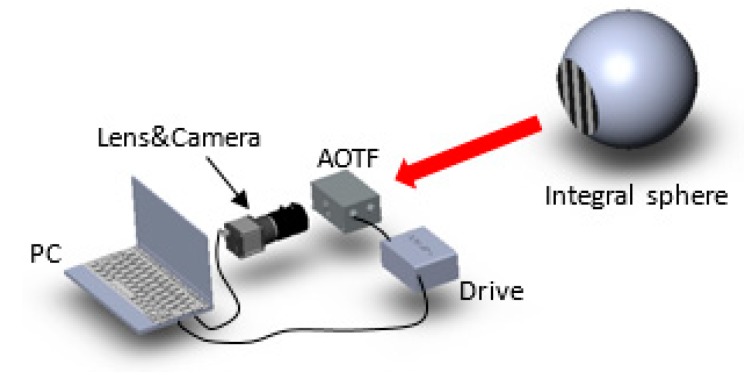
The experiment setup of the camera response contrast. The cut-off wavelength of the AOTF was 1319 nm and the AOTF aperture was the aperture stop in the experiment. The focal lengths of the lenses were 8 mm and 73 mm.

**Figure 8 sensors-19-00390-f008:**
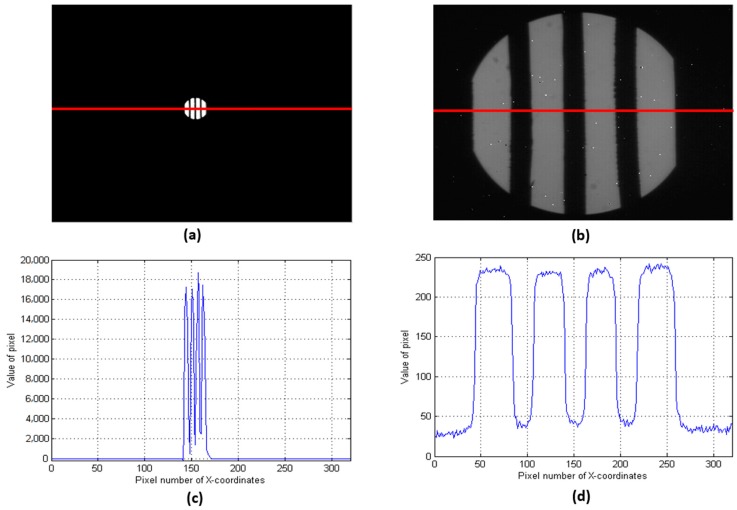
Images of strong light conditions. (**a**) Image acquired by the 8-mm lens, (**b**) image acquired by the 73-mm lens, (**c**) the row of data from the red line shown in (**a**), (**d**) the row of data from the red line shown in (**b**). The data here are the results of 16-bit quantization. For a better display, (**a**,**b**) are not shown in the same grayscale. The response of the 8-mm focus system was larger than that of the 73-mm focus system. However, the images from the 8-mm lens had a low resolution.

**Figure 9 sensors-19-00390-f009:**
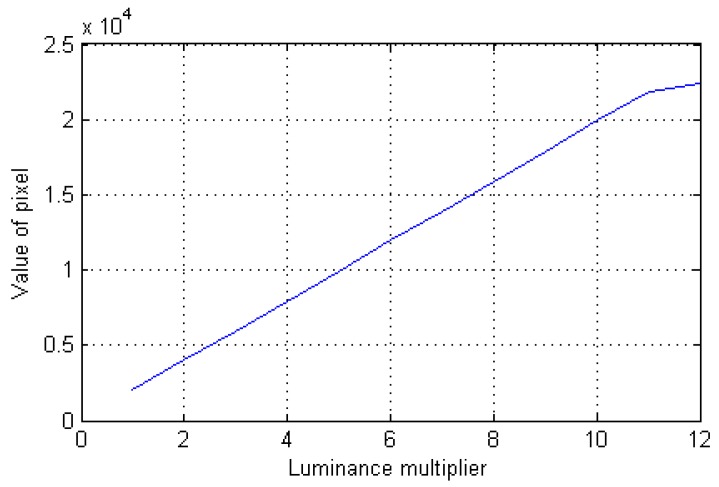
The relationship between the camera response and irradiance.

**Figure 10 sensors-19-00390-f010:**
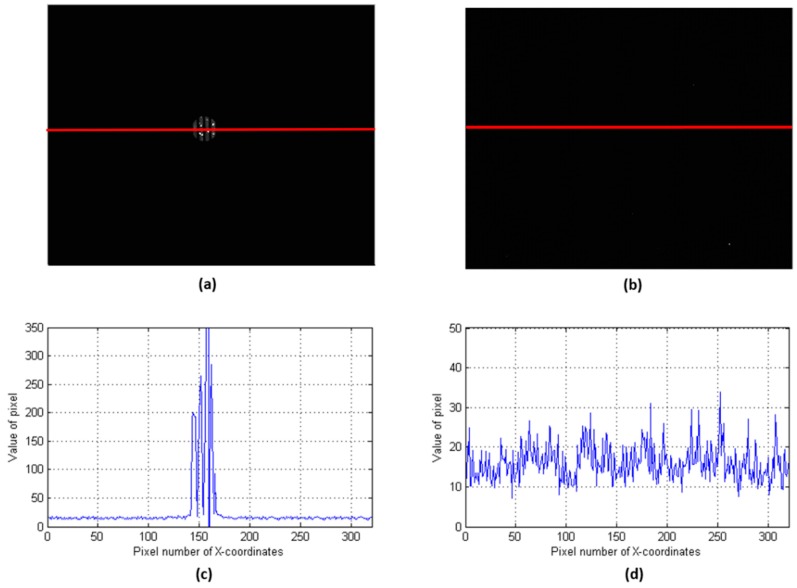
Images of weak light conditions. (**a**) Image acquired by an 8-mm lens, (**b**) image acquired by a 73 mm lens, (**c**) the row of data from the red line shown in (**a**), (**d**) the row of data from the red line shown in (**b**). In weak light conditions, the traditional system was no longer able to obtain a signal.

**Figure 11 sensors-19-00390-f011:**
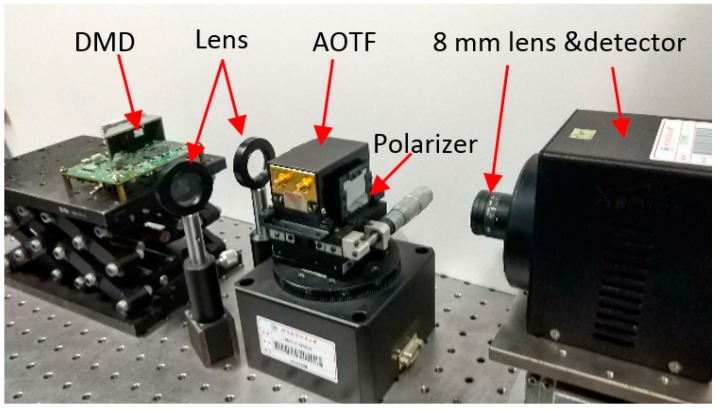
Real conditions of the experimental setup. A polarizer was used to suppress stray light. The lens was focused on the DMD.

**Figure 12 sensors-19-00390-f012:**
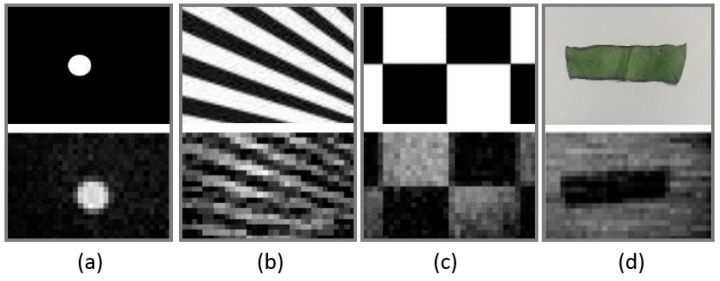
Different target objects and their image results. The objects in (**a**–**d**) correspond to the white circle, oblique stripe, checkerboard, and cut leaf, respectively.

**Figure 13 sensors-19-00390-f013:**
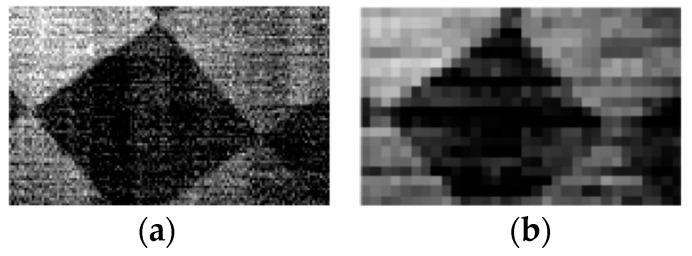
Oblique square images acquired by the SWIR AOTF imaging spectrometer on the basis of SPI. (**a**) High-resolution image, (**b**) low-resolution image.

**Figure 14 sensors-19-00390-f014:**
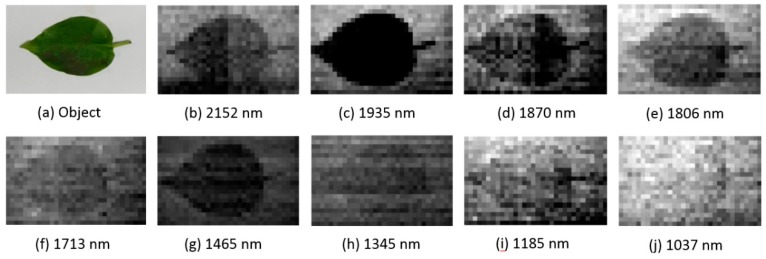
Images with different bands acquired by the SWIR AOTF imaging spectrometer based on SPI. In the 1465 nm and 1935 nm bands, the leaf exhibited less reflectance because H_2_O occupies an absorption peak in these bands.

**Figure 15 sensors-19-00390-f015:**
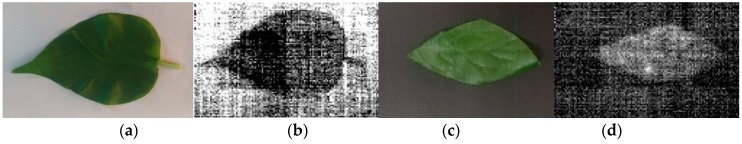
(**a**) Real leaf and (**c**) fake leaf. (**b**,**d**) Images acquired by the imaging spectrometer at 1935 nm. The real leaf is dark and the fake leaf is bright.

**Table 1 sensors-19-00390-t001:** Important parameters of the AOTF and DMD.

**Specification of the AOTF**	
Optical aperture	10 mm × 10 mm
Spectral range	900–2500 nm
Spectral resolution	5–20 nm
Acceptance angle	6–10°
Separation angle	7.1–7.8°
**Specification of the DMD**	
Wavelength range	700–2500 nm
Pattern rate, 8-bit (max)	120 Hz
Micromirror array size	912 × 1140
Micromirror pitch	7.6 µm

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
