# Peer review of "SWIR AOTF Imaging Spectrometer Based on Single-pixel Imaging"

_sensors, 2019, doi:10.3390/s19020390_

Reviewer 1 Report

The paper is well structured and well written. The desriptions, analysis and conclusions are clear. The references formatting should be unified according to the MDPI's requirements.

If the authors write "At present, the AOTF spectrometer is applied to many fields, such as..."

I would expect some citations of the latest papers, while the newest one that was cited is from 2014.

Author Response

Point 1: The references formatting should be unified according to the MDPI's requirements. If the authors write "At present, the AOTF spectrometer is applied to many fields, such as..." I would expect some citations of the latest papers, while the newest one that was cited is from 2014.

Response 1: At first, thank you for your good comments and careful review. The reference formatting has been revised and the citations we are going to add are as follows:

 1.          Marco, C.; Andrea, B.; Stefania, U.; Maurizio, S.; Sonia, E.; Fabio, M.; Rosario, M. On-field monitoring of fruit ripening evolution and quality parameters in olive mutants using a portable NIR-AOTF device. Food Chemistry. 2016, 199, 96-104.

2.          Emmanuel, D.; Jurgen, V.; Bert, V. O.; Didier, F. The AOTF-based NO 2 camera. Atmospheric Measurement Techniques. 2016, 9, 6025-6034.

Reviewer 2 Report

Dear Authors,

In this paper, the authors combined compressed sensing and acousto-optic tunable filter (AOTF) technique to achieve single pixel imaging in the near IR range.

The paper is well organized and the results are sound. I recommend it for publication.

Author Response

Thank you very much for your appreciation.
